# Impact of Malnutrition on Long-Term Mortality in Elderly Patients with Acute Myocardial Infarction

**DOI:** 10.3390/nu11020224

**Published:** 2019-01-22

**Authors:** Klara Komici, Dino Franco Vitale, Angela Mancini, Leonardo Bencivenga, Maddalena Conte, Sandra Provenzano, Fabrizio Vincenzo Grieco, Lucia Visaggi, Ilaria Ronga, Antonio Cittadini, Graziamaria Corbi, Bruno Trimarco, Carmine Morisco, Dario Leosco, Nicola Ferrara, Giuseppe Rengo

**Affiliations:** 1Department of Medicine and Health Sciences, University of Molise, Campobasso, Via Francesco De Sanctis, 1, 86100 Campobasso, Italy; graziamaria.corbi@unimol.it; 2Clinica San Michele, Via Appia 187, 81024 Maddaloni (CE), Italy; dinofranco.vitale@fastwebnet.it; 3Department of Translational Medical Sciences, University of Naples Federico II, Via Pansini 5, 80131 Naples, Italy; angelamancini0387@libero.it (A.M.); leonardobencivenga@gmail.com (L.B.); maddalena-conte@libero.it (M.C.); provenzanosandra@gmail.com (S.P.); fabrizio.g82@hotmail.it (F.V.G.); luciavisaggi@gmail.com (L.V.); ilariaronga87@gmail.com (I.R.); antonio.cittadini@unina.it (A.C.); dleosco@unina.it (D.L.); giuseppe.rengo@unina.it (G.R.); 4Department of Advanced Biomedical Sciences, University of Naples Federico II, Via Pansini, 5, 80131 Naples, Italy; bruno.trimarco@unina.it (B.T.); carmine.morisco@unina.it (C.M.); 5Istituti Clinici Scientifici Maugeri SpA Società Benefit (ICS Maugeri SpA SB), 82037 Telese Terme (BN), Italy

**Keywords:** mini nutritional assessment, acute myocardial infarction, mortality, elderly

## Abstract

Background: Malnutrition is a frequent condition in the elderly, and is associated with prolonged hospitalization and increased mortality. However, the impacts of malnutrition among elderly patients with acute myocardial infarction have not been clarified yet. Methods and Results: We enrolled 174 patients aged 65 years and over, admitted with the diagnosis of acute myocardial infarction (AMI), who underwent evaluation of nutritional status by Mini Nutritional Assessment (MNA) and evaluation of mortality risk by GRACE Score 2.0. All-cause mortality was the outcome considered for this study. Over a mean follow-up of 24.5 ± 18.2 months, 43 deaths have been registered (24.3%). Non-survivors were more likely to be older, with worse glomerular filtration rate, lower systolic blood pressure, lower albumin and MNA score, higher prevalence of Killip classification III-IV grade, and higher Troponin I levels. Multivariate Cox proportional analysis revealed that GRACE Score and MNA showed a significant and independent impact on mortality, (HR = 1.76, 95%, CI = 1.34–2.32, and HR = 0.56, 95% CI = 0.42–0.73, respectively). Moreover, the clinical decision curve revealed a higher clinical net benefit when the MNA was included, compared to the partial models without MNA. Conclusion: Nutritional status is an independent predictor of long-term mortality among elderly patients with AMI. MNA score in elderly patients with AMI may help prognostic stratification and identification of patients with, or at risk of, malnutrition in order to apply interventions to improve nutritional status, and maybe survival in this population.

## 1. Introduction

Malnutrition is a common condition in the elderly, with an overall prevalence higher than 20% [1,2,3,4]. Physical deconditioning, cognitive impairment, and comorbidities with other components of pathophysiological aging such as dysphagia, malabsorption, and derangement of hormonal systems, lead to an imbalance of anabolic–catabolic metabolism that translates in malnutrition [5]. The Mini Nutritional Assessment (MNA) is a validated test recommended for nutritional screening in the geriatric population, and is widely used in different clinical settings including acute care, rehabilitation, nursing home, and community [6]. Malnutrition has been associated with impaired cardio-respiratory performance, increased risk of falls, prolonged hospitalization, and increased mortality [7,8,9]. Importantly, nutritional status evaluated by MNA scale has been demonstrated to predict mortality in general free-living population [10]. The prevalence and the severity of coronary vascular disease increase with age, and more than 35% of patients with acute myocardial infarction (AMI) are ≥75 years old [11,12]. The GRACE risk Score 2.0 algorithm estimates the short and long-term mortality risk in acute coronary syndromes (ACS), and it is mainly based on a variety of cardiovascular parameters, such as heart rate, systolic blood pressure, electrocardiography modifications, cardiac ischemia biomarkers and cardiogenic shock [13].

Recent studies have reported that low body mass index (BMI) and hypoalbuminemia are important risk factors for mortality in elderly patients after ACS [14,15]. Furthermore, evaluation of nutritional status by body weight and albumin level has demonstrated to influence clinical outcomes during hospitalization in patients with myocardial infarction [16]. However, there is a lack of knowledge regarding the impact of nutritional status on long-term outcome, especially in elderly patients affected by acute cardiovascular events, such as AMI. Therefore, we sought to investigate the prognostic value of nutritional status, as detected by MNA, independently from other confounders, on long-term mortality in elderly patients affected by AMI using a multivariable approach in a prospective observational study.

## 2. Materials and Methods

### 2.1. Study Population

All patients aged 65 years and older admitted at the Intensive Coronary Care Unit (ICCU) of the University of Naples Federico II, with the diagnosis of AMI were screened for inclusion in this study. Inclusion criteria were: (a) diagnosis of ST elevated myocardial infarction (STEMI) or non-ST elevated myocardial infraction (NSTEMI); (b) age ≥65 years; (c) willingness to participate in this study. Patients underwent a complete clinical examination and a structured interview about cardiovascular risk factors and other comorbidities. Based on clinical signs of acute heart failure (HF), Killip classification has been also determined to evaluate the severity of HF in patients with AMI. Killip class I identified patients without HF, class II patients with mild HF, class III patients with pulmonary edema and, class IV patients in cardiogenic shock [17]. Baseline laboratory, echocardiography and angiography findings including: White blood cells, hemoglobin, Troponin I, creatine kinase MB isoenzymes, albumin, serum creatinine levels to estimate glomerular filtration rate (GFR), left ventricular ejection fraction (LVEF), severity of coronary artery disease and primary coronary intervention (PCI) were recorded. GRACE risk Score 2.0 for mortality was calculated based on age, heart rate, systolic blood pressure, creatinine and Troponin level, Killip Class, ST segment modifications, and cardiac arrest, as previously described [18]. The study has been approved by the Local Ethical Committee (protocol number 50/13), all patients were informed, and they signed a written consent to participate in this study.

### 2.2. Diagnosis of AMI

AMI was diagnosed according to the third universal definition of myocardial infarction criteria recommend by European Society of Cardiology, the American Heart Association and WHO [19,20]. The diagnostic criteria were as follows: (a) Typical chest pain, upper extremity, mandibular or epigastric discomfort of more than 20 minutes, (b) ST depression ≥ 0.05 mV in at least two contiguous leads and/or T wave inversion ≥ 0.1 mV in at least two contiguous leads with R to S ratio > 1; ST elevation at the J point ≥ 0.1 mV in two contiguous leads except V2-V3 ≥ 0.2 mV in men > 40 years and ≥ 0.15 mV in women, (c) serum Troponin I or creatine kinase levels above the 99th percentile of the normal reference population during the first 24 h after admission.

### 2.3. Mini Nutritional Assessment (MNA)

Nutritional status was assessed with the complete MNA administered by a clinical researcher after the angiography or revascularization procedures. MNA consists of 18 items: Measurements of BMI, calf and mid-arm circumference, questions on food and beverages intake, weight loss, acute disease, mobility, psychological stress, depression, dementia, independent living, polypharmacy, pressure injuries, and self-view of nutritional and general clinical status. It has been reported that a score of ≥ 24 identified a condition of normal nutrition, a score of ≥17 and ≤23.5 identified a condition of malnutrition risk, and a score of <17 indicates malnutrition [21].

### 2.4. Follow-Up

All-cause mortality was the outcome considered for this study. Periodical follow-up was performed by telephone to determine patients’ survival. The final survival status of the population was obtained in May 2017. Follow-up period was terminated on 30 May, 2017, or in case of death.

### 2.5. Statistical Analysis

Continuous variables were expressed as mean ± standard deviation (SD), and compared by the Student’s t test. Categorical variables were expressed as proportion, and compared by the χ^2^ test. The Cox proportional hazard analysis was used to identify the factors associated with all-cause mortality. Considering that GRACE Score included multiple cardiovascular parameters and used parsimonious criteria, we tested the following as potential variables associated with mortality in the Cox model: Age, gender, BMI, LVEF, GRACE Score, diabetes mellitus (DM), albumin level, and MNA score. To determine the contribution of each variable to the global outcome, the partial contribution of each variable to the global R^2^ has been measured. The decision curve analysis [22,23,24,25], a suitable method to test alternative prognostic strategies, was used to estimate the potential additive clinical value of MNA evaluation. Using this method, the net clinical benefit obtained using a new prognostic model has been compared to the other two possible alternatives, i.e., the “treat all” and “treat none” strategies, and the standard prognostic model.

To have a comprehensive view of the relative weight of each factor on the survival curve, the Hazard Ratio (HR) of the significant continuous variables included in the final Cox models were related to one SD variation for both MNA and mean GRACE Score. Directly adjusted survival plots [26], obtained by averaging the Cox survival curves adjusted by the covariate values of each subject, were used to compare the survival at specific factors’ values to the overall Kaplan–Meier. Data were analyzed by Stata version 13.0 (StataCorp LP, College Station, TX, USA). Statistical significance was accepted at *p* ≤ 0.05.

## 3. Results

### 3.1. Patient Characteristics

The final study population consisted of 174 patients, with a mean age of 74.26 ± 7.08 years. 65.0% were males (114/174), with a mean LVEF of 39.89 ± 8.49%. 52.9% had STEMI, and mean GRACE Score of 150.01 ± 24.38. Of the potentially eligible patients that satisfied the inclusion criteria, MNA was not assessed in six patients, while the other seven patients were lost at follow-up. Demographic data of the overall study population, sorted as survivors or non-survivors, are reported in Table 1. Interestingly, no differences were observed between survivors and non-survivors in gender, BMI, presence of diabetes mellitus (DM) or chronic obstructive pulmonary disease (COPD), hypertension, heart rate, or hemoglobin and glycaemia level. However, non-survivors were more likely to be older, with worse GFR (at the edge of the significant threshold, *p* = 0.078 and 0.080, respectively), lower systolic blood pressure (SBP), lower albumin, lower MNA score, higher prevalence of Killip classification III-IV, and higher Troponin I levels. Based on MNA score, 12.1% (21/174) patients were malnourished (MNA < 17), 38.5% (67/174) were at risk for malnutrition (MNA ≥ 17 and ≤ 23.5), and 49.4% (86/174) presented a good nutritional status (MNA ≥ 24). As shown in Appendix A, we have also performed a comparison between the distribution of the baseline characteristics of patients with an MNA score of <24 and MNA score of ≥24. Patients with an MNA score ≥24 had a significant higher body weight (*p* < 0.05), and higher albumin levels (just above the significant threshold, *p* = 0.080).

### 3.2. Follow-Up Analysis

Over a mean follow-up of 24.5 ± 18.2 months, 43 deaths were registered (24.7%). A significant (*p* < 0.001) proportion of non-survivors, 74.4% (32/43), were malnourished or at risk of malnutrition. Mortality rate was significantly higher in the MNA < 24 group (36.4 %; 32/88), compared to the MNA ≥24 group (12.8%; 11/86; *p* < 0.001). Cox analysis revealed that the GRACE Score and MNA were the only significant and independent predictors of mortality, as documented by the HR relative to one standard deviation (SD) variation that was 1.76 (95% CI 1.34–2.32) and 0.56 (95% CI 0.42–0.73), respectively (Table 2). The multivariate model showed a global R^2^ of 34.50%, suggesting that good fraction of the outcome variability was predicted. Interestingly, the contribution of the GRACE and MNA scores to the variation of the prognostic index, as documented by the partial R^2^, was 16.7% and 17.8%, respectively. This latter finding indicates that the nutritional status has an impact on the survival of AMI patients that is comparable to that one exerted by the GRACE Score. The graphical impact of MNA on survival is shown in Figure 1A, where the survival curves relative to ±1 SD variation of MNA mean score are reported along with the overall Kaplan–Meier curve. Survival curves relative to ±1 SD variation of the mean GRACE Score and the overall Kaplan–Meier curve are reported in Figure 1B. These curves have been generated from the Cox parameters, and compared with the overall Kaplan–Meier curve. They represent the survivals that would be observed if the whole population is the given MNA and GRACE Score.

The clinical net benefit which profiles achievable, clinical setting application of the models with and without the MNA score are plotted in Figure 2. Both curves are higher than the net benefit obtained by applying the other two possible approaches, namely the “treat none” and “treat all” strategies. Of note, the net benefit of the full model including the MNA score draws higher than that of the partial model for a large and significant portion of the decision threshold.

## 4. Discussion

The main finding of the present study is that the nutritional status, evaluated by the MNA score, is an independent predictor of long-term mortality in elderly patients with AMI. Indeed, lower MNA score was associated with higher incidence of all-cause mortality. Moreover, the prognostic information obtained by MNA was comparable with that obtained by GRACE Score, as well as by other established prognostic score for long and short-term mortality in AMI patients. Finally, as demonstrated by the decision curve analysis, the introduction of MNA evaluation in the clinical setting of the acute coronary care units might have a beneficial role in identifying and managing patients with worse prognoses.

Several authors have described MNA as an independent predictor of mortality in different settings, such as free-living women [10], and after hip fracture surgery [27,28]. Although, our study population was focused on elderly with AMI, from our data results it is evident that MNA is an independent predictor of mortality. We used specific statistic models to test the impact of malnutrition on mortality, and to understand its specific relationship to survival outcome. In all our models MNA demonstrated to have an important impact on predicting mortality similar to specific cardiovascular characteristics. Recently, different studies have evaluated the role of nutritional status in patients undergoing PCI [29,30,31], indicating that nutritional status has a predictive value on outcome. Yoo et al. [16], in a similar setting, described that undernutrition has an important influence on clinical outcome of post myocardial infarction patients during hospitalization. However, in abovementioned studies, the nutritional status of the elderly population has been evaluated with different scales such as prognostic nutritional index, geriatric nutritional index, and controlling nutritional status score that are generally based on albumin, lymphocytes, and cholesterol levels, or body weight. Consistently, our univariate analysis indicated that albumin level is a significant predictor of mortality. However, in the multivariate analysis, it had less impact on mortality. The presence of more powerful variables, such as GRACE Score, might have masked the prognostic role of albumin in our statistical model.

GRACE Scoring system has been reported as one of the most robust risk stratification score in the clinical practice for patients with ACS [32]. Indeed, in our population, GRACE Score and its main components showed a significant predictive value. In our statistical model, the MNA score, evaluated together with the GRACE Score, showed a predictive role, indicating an independent prognostic impact on long-term mortality.

As expected, LVEF was significantly reduced in non-survivors compared to survivors in univariate analysis. Previous studies by us and others reported LVEF as an independent predictor of long-term mortality in the clinical setting of chronic HF [33,34]. However, in the present study, our multivariate analysis did not reveal a significant predictive role of LVEF. In this regard, it is important to underline that we evaluated LVEF at the time of the acute coronary event, and EF evaluations late after the ACS might represent a better prognostic variable for long-term mortality.

Buchloz et al. [14] reported that low BMI is an independent risk factor for mortality after AMI. In our population, survivors and non-survivors were homogenous for BMI, and multivariate analysis did not report any significant role of BMI.

Some studies have reported an association between increased levels of inflammation biomarkers such as TNF-α, malnutrition status, and poor outcome in patients with HF [35,36,37]. Mechanistically, the activation of neurohormonal and inflammatory pathways that characterize cardiovascular disease may increase the catabolic demand, and patients with already poor nutritional status may be more vulnerable to cardiac events. Moreover, the protective and beneficial role of physical activity on ischemic cardiac disease is well recognized [38]. The link between malnutrition and physical decondition is also well-documented [39]. Thus, it is plausible to assume that poor nutritional status, by limiting the physical performance of post-ischemic elderly patients, may influence their outcome. However, highlighting the mechanisms by which malnutrition affects mortality in ACS patients is beyond the purposes of the present study.

It is well-known that nutritional support can correct the nutritional deficit, and as reported from several studies, supplemental strategies reduce the in-hospital mortality [40,41,42]. The results of our study strongly suggest that besides overall cardiovascular intervention strategies, elderly patients may benefit from geriatric evaluations including investigation of the nutritional status.

### Study Limitation

This study is a single center experience in a relative small number of patients. Thus, our findings deserve confirmation in larger multi-centric studies. Further, socioeconomical factors, comorbidities such as dementia or mild cognitive impairment, mood disorders and poly-therapy may have influenced the impact of malnutrition on outcome. However, in the elderly population, most of these conditions may develop or become clinically evident after an acute event. In regard to this, periodic clinical evaluation after discharge would be necessary to test the impact of these factors on the prognostic value of nutritional status in elderly patients with AMI. The results of our study did not influence nutritional interventions during hospitalization or discharge. Moreover, we did not evaluate if a therapeutic intervention aiming to correct the nutritional status would affect mortality in our study population. Future studies are needed to clarify whether correction of the nutritional status would improve prognosis in elderly patients with AMI.

## 5. Conclusions

Nutritional status is an independent predictor of long-term mortality among elderly patients with AMI. Measurement of MNA score in elderly patients with AMI may help prognostic stratification and identification of patients with, or at risk of, malnutrition in order to apply interventions to improve nutritional status.

## Figures and Tables

**Figure 1 nutrients-11-00224-f001:**
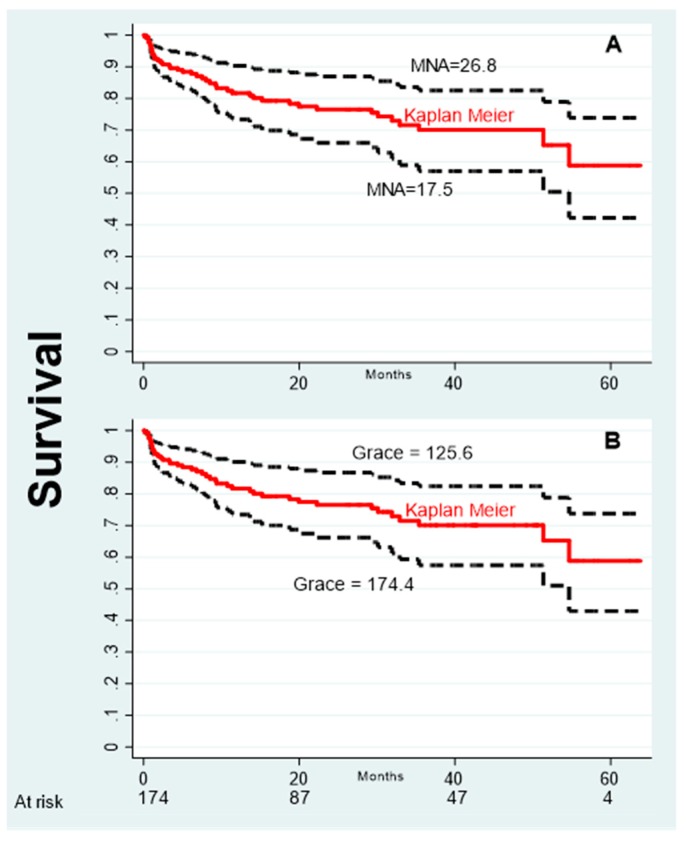
(**A**). Survival curves (dashed black lines) at specific values of mini nutritional assessment (MNA) (mean ± 1 SD) obtained from the Cox model adjusting for the GRACE Score at the values observed in the study population (directly adjusted curves). (**B**) Survival curves (dashed black lines) at specific values of GRACE Score (mean ± 1 SD) obtained from the Cox model adjusting for the MNA score at the values observed in the study population. KM, overall Kaplan–Meier curve (continuous red line).

**Figure 2 nutrients-11-00224-f002:**
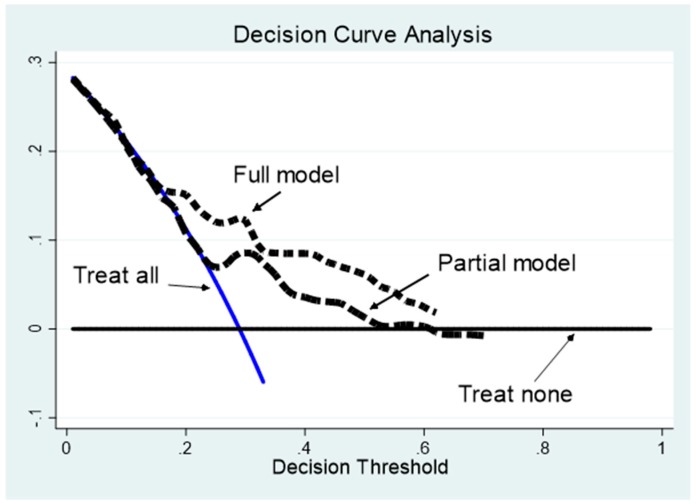
Decision curve analysis. The “treat none” (black continuous line) and “treat all’ (blue continuous line) curves are compared with the net benefit curves of the two Cox models: Full model (short dash black line) and partial model without MNA (long dash black line). The full model profile is higher than the partial model profile across the critical range of the decision threshold probabilities (20–60%). Both models show curves above that of the “treat none” and “treat all” strategies.

**Table 1 nutrients-11-00224-t001:** Characteristic of patients in the overall study population, stratified as survivors and non-survivors.

Population Characteristics	All (174)	Survivors (*N* = 131)	Non-survivors (*N* = 43)	*p*-Value
Age, years ± SD	74.26 ± 7.08	73.73 ± 7.16	75.86 ± 6.65	0.078
Gender, male (*n*, %)	114 (65)	86 (65.6)	28 (65.1)	0.544
BMI, kg/m^2^ ± SD	27.56 ± 5.58	27.39 ± 5.03	28.03 ± 7.02	0.582
LVEF, % ± SD	39.89 ± 8.49	40.96 ± 7.59	36.67 ± 10.17	0.014
Heart Rate bpm, ± SD	78.56 ± 16.77	77.98 ± 15.94	80.33 ± 19.18	0.473
SBP, mmHg ± SD	128.70 ± 23.12	131.26 ± 22.52	120.88 ± 23.42	0.013
STEMI, (*n*, %)	92 (52.9)	69 (52.6)	23 (53.4)	0.532
Killip Class (III, IV *n,* %)	41 (23.5)	15 (11.4)	26 (60.4)	<0.0001
GRACE Score, ± SD	150.01 ± 24.38	146.34 ± 22.48	161.45 ± 26.75	<0.01
MNA, ± SD	22.15 ± 4.67	22.81 ± 4.45	20.13 ± 4.83	<0.01
DM, (*n*, %)	62 (35.6)	45 (34.3)	17 (39.5)	0.354
Hypertension, (*n*, %)	127 (73)	99 (75.5)	28 (65.1)	0.127
Smokers, (*n*, %)	78 (44.8)	59 (45.0)	19(44.1)	0.352
COPD, (*n*, %)	38 (21.8)	27 (20.6)	11 (25.5)	0.537
Hemoglobin, mg/dl ± SD	13.05 ± 1.89	13.12 ± 1.85	12.84 ± 1.99	0.42
WBC × 1000 /µl, ± SD	10.37 ± 3.25	10.14 ± 3.82	11.08 ± 3.87	0.111
Glycemia, mg/dl, ± SD	135.7 ± 53.57	133.0 ± 43.48	143.93 ± 66.82	0.326
GFR ml/kg/m^2^, ± SD	72.38 ± 28.38	74.71 ± 27.18	65.12 ± 31.42	0.080
Albumin mg/dl, ± SD	3.72 ± 0.62	3.78 ± 0.64	3.54 ± 0.52	0.013
Troponin I ng/ml, ± SD	20.54 ± 23.62	14.08 ± 12.11	40.27 ± 36.28	<0.0001
Statins, (*n*, %)	168(97.1)	128 (97.7)	40 (95.2)	0.569
ASA, (*n*, %)	170 (97.7)	130 (98.6)	40 (95.2)	0.248
Beta-blockers, (*n*, %)	137 (78.7)	106 (81)	31 (72.5)	0.315
ACEi/ARBs, (*n*, %)	112 (64.3)	85 (64.9)	27 (61.9)	0.716

Variables are expressed as mean ± SD, and binary data as percentage. *p* value refers to the survivors/non-survivors comparisons. ACE I indicates angiotensin-converting-enzyme inhibitor; ARBs, angiotensin receptor blockers; ASA, acetylsalicylic acid; BMI, body mass index; DM, diabetes mellitus; COPD, chronic obstructive pulmonary disease; GFR, glomerular filtration rate; LVEF, left ventricular ejection fraction; MNA, Mini Nutritional Assessment; SBP, systolic blood pressure; STEMI, ST-elevated myocardial infarction; WBC, white blood cells.

**Table 2 nutrients-11-00224-t002:** Multivariable Cox proportional hazard regression model.

Independent Variables	HR (95% CI)	*p*-Value	Global R^2^ = 34.50% Fraction R^2^
Age	1.02 (0.98–1.07)	0.265	NA
Gender	1.15 (0.52–2.55)	0.723	NA
BMI	1.01 (0.96–1.06)	0.536	NA
LVEF	0.96 (0.93–1.01)	0.089	NA
DM	1.50 (0.78–2.90)	0.221	NA
MNA + 1 SD	0.56 (0.42–0.73)	< 0.0001	16.70%
Albumin	0.68 (0.39–3.31)	0.221	NA
GRACE Score +1 SD	1.76 (1.34–2.32)	< 0.0001	17.80%

HR, hazard ratio; CI, confidential interval; SD, standard deviation; LVEF, left ventricular ejection fraction; MNA, mini nutritional assessment; DM, diabetes mellitus; BMI, body mass index

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
