# Peer review of "Impact of Malnutrition on Long-Term Mortality in Elderly Patients with Acute Myocardial Infarction"

_nutrients, 2019, doi:10.3390/nu11020224_

Reviewer 1 Report

The manuscript attempts to study the effect of malnutrition, as measured with MNA, on survival of patients admitted due to AMI. In this small sample, the authors show that both the GRACE and MNA scores have equal and significant effects on all-cause mortality among elderly individuals included.

I have several comments

The authors use elaborate statistical methods to prove the effect of MNA on survival. This is very hard to follow and understand. I could not understand figure 2, and nor do I think it has bearing on the actual conclusions. The authors may consider omitting it, or perhaps attaching it as a supplementary figure.

I would like the authors to provide the reason for choosing this elaborate statistical analysis.

I would like to see univariate analysis for survival across MNA sub-groups as well. The authors should also provide an analysis of BMI, albumin and cholesterol across MNA sub-groups. This should serve as a validation to their results.

Minor comments

Number of patients do not add up – 1 is missing (86, 62 and 25 for well nourished, at risk and malnourished respectively).

The authors should add information whether the malnourished patients were treated with nutritional intervention, and whether this was recommended in the discharge letters.   

Author Response

The authors use elaborate statistical methods to prove the effect of MNA on survival. This is very hard to follow and understand. I could not understand figure 2, and nor do I think it has bearing on the actual conclusions. The authors may consider omitting it, or perhaps attaching it as a supplementary figure. I would like the authors to provide the reason for choosing this elaborate statistical analysis.

We want to thank this Reviewer for giving us the opportunity to better explain the statistical analysis employed in the present manuscript, which is finalized to accomplish the following aims:

1) to assess the prognostic relevance of the nutritional status by the MNA, on the survival of elderly patients with acute myocardial infarction.

2) to evaluate the independent effect of MNA on survival,  by comparing its relevance with other variables that are known to impact the survival of patients in this clinical setting.

3) to estimate the clinical benefit of MNA assessment once it is added to other variables (known to be relevant in this clinical setting) in the clinical decision-making process.

In order to accomplish Aims 1 and 2, the Cox proportional multivariable analysis has been employed followed by the partition over the significant factors of the overall explained outcome variability (R2). We share the view that “… Advances in statistical methodology now allow us to create and estimate more realistic models than ever before, and the necessary computer programs are often available…” (1). The choice of a multivariable approach has been mandatory to evaluate the potential prognostic power of MNA independent from the other variables, considered as potential confounders.

It should be noted that the use of a univariate analysis may not be adequate and can result misleading, as explained by Royston (2). In the Rotterdam Breast Cancer Dataset this Authors showed that hormone therapy exhibits a significant harmful effect on survival at univariate analysis (Hazard Ratio 1.33, 95% CI 1.08-1.63); at the opposite, the multivariate analysis revealed that the hormone therapy results in a significant beneficial effect (after correction for the other confounding variables - Hazard Ratio 0.51, 95% CI 0.41-0.65).

Regarding Aim 3, translation of a prognostic model into clinical practice is the final goal of a clinical investigation. Considering the results derived from the dataset used to develop the prognostic model, the statistical significance does not translate de facto into clinical utility. Indeed, the AUC metric derived from ROC analysis, albeit commonly used, is not a satisfactory measure of clinical utility (3). Importantly, taking advantage from the results of the studies by Vickers (3-6), it is now possible to have an estimation of the clinical benefit obtained from the same dataset used to develop the prognostic model. This procedure (named Decision Curve Analysis) measures the net (cost/benefit balanced) clinical benefit of applying a given prognostic factor in the medical decision-making process.

Therefore, to accomplish the 3rd aim of the present study, the decision curve analysis has been used to estimate whether the employment of MNA score in the clinical management of AMI patients could add a net clinical benefit over the other alternative clinical approaches (namely: manage patients with the standard procedure [not considering MNA score], manage all patients indiscriminately [‘treat all’] or ignore all prognostic models and do not manage patients [‘treat none’].

Based on Reviewer comment, the following changes have been performed in the revised version of our manuscript.

Introduction section, the last paragraph  has been changed as follows:

“However, there is a lack of knowledge regarding the impact of nutritional status on long term outcome, especially in elderly patients affected by acute cardiovascular events, such as AMI. Therefore, we sought to investigate the prognostic value of nutritional status, as detected by MNA, independently from other confounders, on long-term mortality in elderly patients affected by AMI using a multivariable approach in a prospective observational study”.

To improve the literature representation of the Decision Curve Analysis the following 3 references have been added  (in the text reference 23-25).

i)             Vickers AJ, Elkin EB. Decision curve analysis: a novel method for evaluating prediction models. Med Decis Making. 2006 Nov-Dec;26(6):565-74.

ii)            Vickers AJ. Decision analysis for the evaluation of diagnostic tests, prediction models and molecular markers. Am Stat. 2008;62(4):314-320.

iii)           Vickers AJ, Cronin AM, Elkin EB, Gonen M. Extensions to decision curve analysis, a novel method for evaluating diagnostic tests, prediction models and molecular markers. BMC Med Inform Decis Mak. 2008 Nov 26;8:53.

The paragraph starting with “ With this method…”   of the Statistical Analysis section has been changed as follow:

“With this method, the net clinical benefit obtained using a new prognostic model has been compared to the other 2 possible alternatives, i.e. the “treat all” and “treat none” strategies and to the standard prognostic model.”

Answer References

1)            Royston P, Sauerbrei W. Multivariate Model Building. A Pragmatic Approach to Regression Analysis Based on Fractional Polynomials for Modeling Continuous Variables. Chichester, United Kingdom: Wiley; 2008. pp 18-19.

2)            Royston P, Lambert P C. Flexible Parametric Survival Analysis Using Stata: Beyond the Cox Model. College Station, Texas USA: Stata Press 2011. pp 143-145.

3)            Vickers AJ. Decision analysis for the evaluation of diagnostic tests, prediction models and molecular markers. Am Stat. 2008;62(4):314-320.

4)            Vickers AJ, Elkin EB. Decision curve analysis: a novel method for evaluating prediction models. Med Decis Making. 2006 Nov-Dec;26(6):565-74.

5)            Vickers AJ, Cronin AM, Elkin EB, Gonen M. Extensions to decision curve analysis, a novel method for evaluating diagnostic tests, prediction models and molecular markers. BMC Med Inform Decis Mak. 2008 Nov 26;8:53.

6)            Van Calster B, Vickers A.J. Calibration of Risk Prediction Models:Impact on Decision-Analytic Performance. Med Decis Making. 2015 Feb;35(2):162-9.

I would like to see univariate analysis for survival across MNA sub-groups as well. The authors should also provide an analysis of BMI, albumin and cholesterol across MNA sub-groups. This should serve as a validation to their results.

According to the Reviewer requests, we have now performed univariate analysis between two MNA sub-groups: patients with MNA score < 24 (patients with malnutrition status and at risk for malnutrition) vs. patients with MNA score≥ 24 (patients with a good nutritional status). In this way the distribution of the population was more balanced 88 vs 86 patients. The univariate analysis showed a significant reduced survival in patients with a MNA score < 24 compared to patients with MNA ≥ 24 pts. (36.4 % vs 12.8 % p<0.001). In the revised version of our manuscript, we have provided also the main baseline characteristics of these two MNA sub-groups, including BMI, albumin and cholesterol levels). In our population, patients with MNA≥ 24 showed higher body weight and are more likely to have higher albumin levels (at the edge of the significance threshold). For other baseline characteristics, we did not find significant differences. We have introduced these data in the new supplementary table 1. Accordingly, in the main text the following statement has been added:  “As shown in Supplemental Table 1, we have also performed a comparisons between the distribution of the baseline characteristics of patients with an MNA score < 24 and MNA score≥ 24. Patients with an MNA score≥ 24 had a significant higher body weight (p <0.05) and higher albumin levels (just above the significant threshold p=0.080).

Moreover, in the follow-up section, we have now added the following statement: “A significant (p<0.001) proportion of non-survivals 74.4% (32/43) were malnourished or at risk for malnutrition. Mortality rate was significantly higher in the MNA <24 group (36.4 %; 32/88), compared to the MNA≥ 24 group (12.8%; 11/86; p < 0.001)”.

Minor comments

Number of patients do not add up – 1 is missing (86, 62 and 25 for well nourished, at risk and malnourished respectively).

We want to thank the Reviewer to raise this inaccuracy to our attention and we apologies for the mistake. Indeed, using the criteria of MNA≤17 (malnourished), MNA>17&MNA≤23.5 (at risk of malnutrition) and MNA≥24 (well nourished) classifies 25, 63 (not 62) and 86 subjects in the 3 classes, respectively. However, making this correction we realized that the criteria used to make this computation were slightly different from those proposed by Guigoz Y et al. (1) (despite in the original version of our manuscript, the criteria had been correctly reported). Indeed, based on the criteria by Guigzon and colleagues (1), the first class (malnourished) threshold is MNA<17 17="" not="" and="" the="" second-class="" at="" risk="" of="" threshold="" is="" mna="">17). Accordingly, we have recalculated the number of patients in the 3 MNA sub-groups, and the text has been modified as follow:“Based on MNA score, 12.1% (21/174) patients were malnourished (MNA< 17), 38.5% (67/174) were at risk for malnutrition (MNA≥17 and ≤23.5) and 49.4% (86/174) presented a good nutritional status (MNA≥24).”

1.  Guigoz Y VB, Garry PJ. Mini nutritonal assessment: a practical assessment tool for grading the nutritional state of elderly patients. Facts Res Gerontol 1994; 4: 15–59

The authors should add information whether the malnourished patients were treated with nutritional intervention, and whether this was recommended in the discharge letters.  

Bases on the observational nature of our study, nutritional interventions were beyond the aims of the present investigation. Indeed, the results of our study did not influence specific nutritional interventions. In the revised version of our manuscript we have now added the following statement in study limitations section: “The results of our study did not influence nutritional interventions during hospitalization or discharge”.

Reviewer 2 Report

Thank you for the opportunity to review the manuscript: "Impact of malnutrition on long-term mortality in elderly patients with acute myocardial infarction." The topic is significant for health care professionals and could have several consequences for people.  

The paper itself is written understandably and sounds appropriate, but some clarifications are needed.

The abstract section contains some acronyms that must be clarified even if familiar to the most, in example row 27.

Introduction: the first reference is aged, do the Authors can find some more recent evidence about malnutrition condition in old people?  i.e. Wolters M et al., Prevalence of malnutrition using harmonised definitions in older adults from different settings e A MaNuEL study, Clinical Nutrition, https://doi.org/10.1016/j.clnu.2018.10.020

Row 60 the Aim of the study is too simple to the discussion and need a better explanation, I would prefer to read also the research question. Maybe would be useful to describe study design even.

Material and Methods: due to the Italian regulation law, Ethical Committee should approve the study explicitly, so please report the approval registration number.

Mini Nutritional Assessment is not validated in the Italian language, how Authors deal with this limit?

Authors also need to describe when patients have been contacted and the overall procedure to achieve the data about mortality.

Results: the final study population reported was 174 patients, which was the eligibility rate and the adhesion rate to the follow-up?. Row 131-133, Authors declared a significant proportion of non-survivals patients were malnourished or at risk of malnutrition; when they assess this event? At the end of the study, or before?

Discussion: row 230, only one study reported, maybe  "several" could be omitted.

Study limitation: Author need to describe how they deal with potential bias as comorbidity, socio-economical issue and others 

Author Response

Reviewer # 2

We would like to thank the Reviewer for the favorable remarks and constructive criticism, which helped us improve the overall quality of our manuscript. In response to the Reviewer’ major and minor concerns we have made changes to the manuscript that are highlighted in red.

The abstract section contains some acronyms that must be clarified even if familiar to the most, in example row 27.

As suggested by this Reviewer, we have clarified the acronyms GFR (glomerular filtration rate) and SBP (systolic blood pressure) in the text section and removed these acronyms from the abstract.

Introduction: the first reference is aged, do the Authors can find some more recent evidence about malnutrition condition in old people?  i.e. Wolters M et al., Prevalence of malnutrition using harmonised definitions in older adults from different settings e A MaNuEL study, Clinical Nutrition, https://doi.org/10.1016/j.clnu.2018.10.020

We want to thank the Reviewer for this suggestion. More recent references (published between 2016-2018) have been identified and added to the bibliography (new references number 2-4):

Rojer AG, Kruizenga HM, Trappenburg MC, Reijnierse EM, Sipila S, Narici MV, et al. The prevalence of malnutrition according to the new ESPEN definition in four diverse populations. Clin Nutr (Edinburgh, Scotland) 2016;35(3):758e62.

Bonetti L, Terzoni S, Lusignani M, Negri M, Froldi M, Destrebecq A. Prevalence of malnutrition among older people in medical and surgical wards in hospital and quality of nutritional care: A multicenter, cross-sectional study. J Clin Nurs. 2017 Dec;26(23-24):5082-5092. doi: 10.1111/jocn.14051. Epub 2017 Sep 29.

Wolters M, Volkert D, Streicher M, Kiesswetter E, Torbahn G, O'Connor EM, O'Keeffe M, Kelly M, O'Herlihy E, O'Toole PW, Timmons S, O'Shea E, Kearney P, van Zwienen-Pot J, Visser M, Maitre I, Van Wymelbeke V, Sulmont-Rossé C, Nagel G, Flechtner-Mors M, Goisser S, Teh R, Hebestreit A; MaNuEL consortium. Prevalence of malnutrition using harmonized definitions in older adults from different settings - A MaNuEL study. Clin Nutr. 2018 Nov 3. pii: S0261-5614(18)32491-9. doi: 10.1016/j.clnu.2018.10.020.

Row 60 the Aim of the study is too simple to the discussion and need a better explanation, I would prefer to read also the research question. Maybe would be useful to describe study design even.

Based on Reviewer’s suggestion, we have modified the text as follows: ” However, there is a lack of knowledge regarding the impact of nutritional status on long term outcome, especially in elderly patients affected by acute cardiovascular events, such as AMI. Therefore, we sought to investigate the prognostic value of nutritional status, as detected by MNA and independently by other confounders, on long-term mortality in elderly patients affected by AMI using a multivariable approach in a prospective observational study.”

Material and Methods: due to the Italian regulation law, Ethical Committee should approve the study explicitly, so please report the approval registration number.

As suggested by the Reviewer, the study approval registration number (protocol number 50/13) has been added in the text.

Mini Nutritional Assessment is not validated in the Italian language, how Authors deal with this limit?

Although the lack of a proper validation study for the MNA Italian version, in the Italian population the MNA score has been used in several studies as part of the multidimensional geriatric assessment Following a list of references where the MNA has been used in its Italian version:

Pilotto A et al. Multidimensional Prognostic Index based on a comprehensive geriatric assessment predicts short-term mortality in older patients with heart failure. Circ Heart Fail. 2010 Jan;3(1):14-20. doi: 10.1161/CIRCHEARTFAILURE.109.865022. Epub 2009 Oct 22.

Sancarlo D et al. Validation of a Modified-Multidimensional Prognostic Index (m-MPI) including the Mini Nutritional Assessment Short-Form (MNA-SF) for the prediction of one-year mortality in hospitalized elderly patients. J Nutr Health Aging. 2011 Mar;15(3):169-73.

Abete P. et al. The Italian version of the "frailty index" based on deficits in health: a validation study. Aging Clin Exp Res. 2017 Oct;29(5):913-926. doi: 10.1007/s40520-017-0793-9. Epub 2017 Jul 7.

The MNA form in Italian and guide to compilation as well are available at: https://www.mna-elderly.com/mna_forms. 

Authors also need to describe when patients have been contacted and the overall procedure to achieve the data about mortality.

Periodical follow-up has been performed by telephone to determine patients’ survival. The final survival status of the population has been obtained in May 2017. Follow-up period was terminated on May 30th 2017 or in the case of death. This statement has been reported in follow-up section.

Results: the final study population reported was 174 patients, which was the eligibility rate and the adhesion rate to the follow-up?. Row 131-133, Authors declared a significant proportion of non-survivals patients were malnourished or at risk of malnutrition; when they assess this event? At the end of the study, or before?

The final study population consisted of 174 patients. In this population both MNA assessment and determination of survival have been performed. From the potentially eligible patients that satisfied the inclusion criteria, MNA has not been assessed in six patients, while other seven patients have been lost at follow-up. This information has been added to the text. As mentioned in the text, MNA was administered to patients at the time of the enrollment (during hospitalization).

Discussion: row 230, only one study reported, maybe  "several" could be omitted.

We have added the following references (reference number 41 and 42): 

Heidegger CP, Berger MM, Graf S, Zingg W, Darmon P, Costanza MC, Thibault R, Pichard C. Optimisation of energy provision with supplemental parenteral nutrition in critically ill patients: a randomised controlled clinical trial. Lancet. 2013;381:385–393. doi: 10.1016/S0140-6736(12)61351-8. 

Kutsogiannis J, Alberda C, Gramlich L, Cahill NE, Wang M, Day AG, Dhaliwal R, Heyland DK. Early use of supplemental parenteral nutrition in critically ill patients: results of an international multicenter observational study. Crit Care Med. 2011;39:2691–2699.

Study limitation: Author need to describe how they deal with potential bias as comorbidity, socio-economical issue and others 

We agree with the Reviewer that the impact of nutritional status on long-term survival could be influenced by other unmeasured conditions, such as: socio-economical factors, comorbidities i.e. dementia or mild cognitive impairment, mood disorders and poly-therapy. Importantly, in the elderly population most of these conditions may also develop or become clinically evident after an acute event (such as AMI). At this regard, periodic clinical evaluation of elderly patients after discharge would be of interest and helpful. We have added the following statement to study limitation section: “Further, socio-economical factors, comorbidities such as dementia or mild cognitive impairment, mood disorders and poly-therapy may have influenced the impact of malnutrition on outcome. However, in the elderly population most of these conditions may develop or become clinically evident after an acute event. At this regard, periodic clinical evaluation after discharge would be necessary to test the impact of these factors on the prognostic value of nutritional status in elderly patients with AMI.”

Reviewer 3 Report

Summary

This is an interesting study discussing the influence of malnutrition in elderly with acute myocardial infarction. The overall quality of this manuscript is very good. The manuscript is carefully and well-written and the graphics and tables add to the presentation.

Broad comments 

I have only minor remarks and suggestions to the authors to improve the manuscript, which already has a good overall quality

·       Use either small lettering or capitals in words like “troponin” and “grace score” and use dash or no dash between words line “all-cause” in the whole document

·       Use either comma or dot in decimal numbers

·       Revise the abbreviations introduced (some not introduced, some only used once or twice and therefore unnecessary)

Specific comments 

·       Line 24: revise language

·       Lines 27: introduce abbreviations

·       Line 50: revise language

·       Line 70: Consider explaining the Killip classification as might be unknown to readers

·       Line 73: introduce “CAD”

·       Line 86: avoid line break between number and unit

·       Line 98:  why was it decided not to end the study at the same time for all patients?

·       Lines 126-128: introduce the abbreviations

·       Line 134: avoid page-breaks inbetween the table, introduce “ASA”

·       Line 145: avoid using HR for both heart rate and hazard ratio

·       Line 162-167: revise font size

·       Line 211: introduce HF

Author Response

Reviewer #3

We would like to thank the Reviewer for the favorable remarks and constructive criticism, which helped us improve the overall quality of our manuscript. In response to the Reviewer’  concerns we have made changes to the manuscript that are highlighted in red.

Use either small lettering or capitals in words like “troponin” and “grace score” and use dash or no dash between words line “all-cause” in the whole document

We want to thank the Reviewer for this suggestion. Accordingly, the words “troponin”, “Grace Score” and ‘’all-cause’’ have been now written in a identical format, as follow: Troponin, Grace Score and all-cause .

Use either comma or dot in decimal numbers

 As suggested, in all decimal numbers we have now used dots.

Revise the abbreviations introduced (some not introduced, some only used once or twice and therefore unnecessary)

The abbreviations not introduced have been now clarified, while, those used only once or twice have been eliminated.

Specific comments

Line 24: revise language

The sentence has been modified as follows: “We have enrolled 174 patients aged 65 years and over, admitted with the diagnosis of acute myocardial infarction (AMI) who underwent  evaluation of nutritional status by Mini Nutritional Assessment (MNA) and evaluation of mortality risk by Grace score Score 2.0.”

Lines 27: introduce abbreviations

Glomerular filtration rate (GFR) and systolic blood pressure (SBP) were introduced.

Line 50: revise language

The sentence has been revised as follows: “Importantly, nutritional status evaluated by MNA scale has demonstrated to predict mortality in general free-living population.”

Line 70: Consider explaining the Killip classification as might be unknown to readers

Killip classification has been now explained in the text and an appropriate reference added. The following sentence has been added: “Based on clinical signs of acute heart failure (HF), Killip classification has been also determined to evaluate the severity of HF in patients with AMI. Killip class I identifies patients without HF, class II patients with mild HF, class III patients with pulmonary edema and, class IV patients in cardiogenic shock.

Reference number 17: Killip T, Kimball JT (1967) Treatment of myocardial infarction in a coronary care unit. A two year experience with 250 patients. Am J Cardiol 20:457–464.

Line 73: introduce “CAD”

Done.

Line 86: avoid line break between number and unit

Corrected according to Reviewer’s suggestion.

Line 98:  why was it decided not to end the study at the same time for all patients?

We apologies if we did not adequately clarify this point. Periodical follow-up has been performed by telephone to determine patients’ survival. The final survival status of the population has been obtained in May 2017. Follow-up period was terminated on May 30th 2017 or in the case of death. This information is now reported in the Method section.

Lines 126-128: introduce the abbreviations

Abbreviations for DM and COPD have been introduced.

Line 134: avoid page-breaks inbetween the table, introduce “ASA”

Done. You can now find table nr.1 on page 4 and, acetylsalicylic acid has been introduced for ASA.

Line 145: avoid using HR for both heart rate and hazard ratio

As suggested, Heart rate has been written without abbreviations.

Line 162-167: revise font size

Done.

Line 211: introduce HF

According to Reviewer’s suggestion on Killip class, HF has been introduced in methods section.

Round  2

Reviewer 1 Report

Nothing to add

Reviewer 2 Report

Than you for allowing me to check the revised version of this paper.

Authors have clarified all the comments suggested in their cover letter, and the manuscript has also been significantly improved.

The topic is exciting, and fundamentals care now under the spotlight worldwide, this paper can suggest more insightful research on nutritional outcomes in the acute setting.